# Deep Knowledge Tracing

**Chris Piech**[*], **Jonathan Bassen**[*], **Jonathan Huang**[*‡], **Surya Ganguli**[*],
**Mehran Sahami**[*], **Leonidas Guibas**[*], **Jascha Sohl-Dickstein**[*†]
[*]Stanford University, [†]Khan Academy, [‡]Google
{piech,jbassen}@cs.stanford.edu, jascha@stanford.edu,

## Abstract

Knowledge tracing—where a machine models the knowledge of a student as they interact with coursework—is a well established problem in computer supported education. Though effectively modeling student knowledge would have high educational impact, the task has many inherent challenges. In this paper we explore the utility of using Recurrent Neural Networks (RNNs) to model student learning. The RNN family of models have important advantages over previous methods in that they do not require the explicit encoding of human domain knowledge, and can capture more complex representations of student knowledge. Using neural networks results in substantial improvements in prediction performance on a range of knowledge tracing datasets. Moreover the learned model can be used for intelligent curriculum design and allows straightforward interpretation and discovery of structure in student tasks. These results suggest a promising new line of research for knowledge tracing and an exemplary application task for RNNs.

## 1 Introduction

Computer-assisted education promises open access to world class instruction and a reduction in the growing cost of learning. We can develop on this promise by building models of large scale student trace data on popular educational platforms such as Khan Academy, Coursera, and EdX.

Knowledge tracing is the task of modelling student knowledge over time so that we can accurately predict how students will perform on future interactions. Improvement on this task means that resources can be suggested to students based on their individual needs, and content which is predicted to be too easy or too hard can be skipped or delayed. Already, hand-tuned intelligent tutoring systems that attempt to tailor content show promising results [28]. One-on-one human tutoring can produce learning gains for the average student on the order of two standard deviations [5] and machine learning solutions could provide these benefits of high quality personalized teaching to anyone in the world for free. The knowledge tracing problem is inherently difficult as human learning is grounded in the complexity of both the human brain and human knowledge. Thus, the use of rich models seems appropriate. However most previous work in education relies on first order Markov models with restricted functional forms.

In this paper we present a formulation that we call Deep Knowledge Tracing (DKT) in which we apply flexible recurrent neural networks that are 'deep' in time to the task of knowledge tracing. This family of models represents latent knowledge state, along with its temporal dynamics, using large vectors of artificial 'neurons', and allows the latent variable representation of student knowledge to be learned from data rather than hard-coded. The main contributions of this work are:

1. A novel way to encode student interactions as input to a recurrent neural network.
2. A 25% gain in AUC over the best previous result on a knowledge tracing benchmark.
3. Demonstration that our knowledge tracing model does not need expert annotations.
4. Discovery of exercise influence and generation of improved exercise curricula.

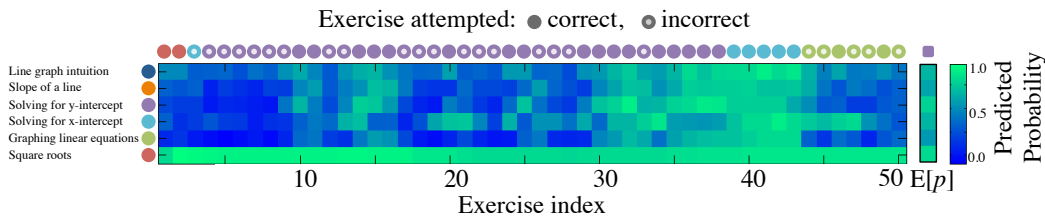

*Figure 1: A single student and her predicted responses as she solves 50 Khan Academy exercises. She seems to master finding x and y intercepts and then has trouble transferring knowledge to graphing linear equations.*

The task of knowledge tracing can be formalized as: given observations of interactions $\mathbf{x}_0 \ldots \mathbf{x}_t$ taken by a student on a particular learning task, predict aspects of their next interaction $\mathbf{x}_{t+1}$ [6]. In the most ubiquitous instantiation of knowledge tracing, interactions take the form of a tuple of $\mathbf{x}_t = \{q_t, a_t\}$ that combines a tag for the exercise being answered $q_t$ with whether or not the exercise was answered correctly $a_t$. When making a prediction, the model is provided the tag of the exercise being answered, $q_t$ and must predict whether the student will get the exercise correct, $a_t$. Figure 1 shows a visualization of tracing knowledge for a single student learning 8th grade math. The student first answers two square root problems correctly and then gets a single x-intercept exercise incorrect. In the subsequent 47 interactions the student solves a series of x-intercept, y-intercept and graphing exercises. Each time the student answers an exercise we can make a prediction as to whether or not she would answer an exercise of each type correctly on her next interaction. In the visualization we only show predictions over time for a relevant subset of exercise types. In most previous work, exercise tags denote the single "concept" that human experts assign to an exercise. Our model can leverage, but does not require, such expert annotation. We demonstrate that in the absence of annotations the model can autonomously learn content substructure.

## 2 Related Work

The task of modelling and predicting how human beings learn is informed by fields as diverse as education, psychology, neuroscience and cognitive science. From a social science perspective learning has been understood to be influenced by complex macro level interactions including affect [21], motivation [10] and even identity [4]. The challenges present are further exposed on the micro level. Learning is fundamentally a reflection of human cognition which is a highly complex process. Two themes in the field of cognitive science that are particularly relevant are theories that the human mind, and its learning process, are recursive [12] and driven by analogy [13].

The problem of knowledge tracing was first posed, and has been heavily studied within the intelligent tutoring community. In the face of aforementioned challenges it has been a primary goal to build models which may not capture all cognitive processes, but are nevertheless useful.

### 2.1 Bayesian Knowledge Tracing

Bayesian Knowledge Tracing (BKT) is the most popular approach for building temporal models of student learning. BKT models a learner's latent knowledge state as a set of binary variables, each of which represents understanding or non-understanding of a single concept [6]. A Hidden Markov Model (HMM) is used to update the probabilities across each of these binary variables, as a learner answers exercises of a given concept correctly or incorrectly. The original model formulation assumed that once a skill is learned it is never forgotten. Recent extensions to this model include contextualization of guessing and slipping estimates [7], estimating prior knowledge for individual learners [33], and estimating problem difficulty [23].

With or without such extensions, Knowledge Tracing suffers from several difficulties. First, the binary representation of student understanding may be unrealistic. Second, the meaning of the hidden variables and their mappings onto exercises can be ambiguous, rarely meeting the model's expectation of a single concept per exercise. Several techniques have been developed to create and refine concept categories and concept-exercise mappings. The current gold standard, Cognitive Task Analysis [31] is an arduous and iterative process where domain experts ask learners to talk through

their thought processes while solving problems. Finally, the binary response data used to model transitions imposes a limit on the kinds of exercises that can be modeled.

## 2.2 Other Dynamic Probabilistic Models

Partially Observable Markov Decision Processes (POMDPs) have been used to model learner behavior over time, in cases where the learner follows an open-ended path to arrive at a solution [29]. Although POMDPs present an extremely flexible framework, they require exploration of an exponentially large state space. Current implementations are also restricted to a discrete state space, with hard-coded meanings for latent variables. This makes them intractable or inflexible in practice, though they have the potential to overcome both of those limitations.

Simpler models from the Performance Factors Analysis (PFA) framework [24] and Learning Factors Analysis (LFA) framework [3] have shown predictive power comparable to BKT [14]. To obtain better predictive results than with any one model alone, various ensemble methods have been used to combine BKT and PFA [8]. Model combinations supported by AdaBoost, Random Forest, linear regression, logistic regression and a feed-forward neural network were all shown to deliver superior results to BKT and PFA on their own. But because of the learner models they rely on, these ensemble techniques grapple with the same limitations, including a requirement for accurate concept labeling.

Recent work has explored combining Item Response Theory (IRT) models with switched nonlinear Kalman filters [20], as well as with Knowledge Tracing [19, 18]. Though these approaches are promising, at present they are both more restricted in functional form and more expensive (due to inference of latent variables) than the method we present here.

## 2.3 Recurrent Neural Networks

Recurrent neural networks are a family of flexible dynamic models which connect artificial neurons over time. The propagation of information is recursive in that hidden neurons evolve based on both the input to the system and on their previous activation [32]. In contrast to hidden Markov models as they appear in education, which are also dynamic, RNNs have a high dimensional, continuous, representation of latent state. A notable advantage of the richer representation of RNNs is their ability to use information from an input in a prediction at a much later point in time. This is especially true for Long Short Term Memory (LSTM) networks—a popular type of RNN [16].

Recurrent neural networks are competitive or state-of-the-art for several time series tasks–for instance, speech to text [15], translation [22], and image captioning [17]–where large amounts of training data are available. These results suggest that we could be much more successful at tracing student knowledge if we formulated the task as a new application of temporal neural networks.

# 3 Deep Knowledge Tracing

We believe that human learning is governed by many diverse properties – of the material, the context, the timecourse of presentation, and the individual involved – many of which are difficult to quantify relying only on first principles to assign attributes to exercises or structure a graphical model. Here we will apply two different types of RNNs – a vanilla RNN model with sigmoid units and a Long Short Term Memory (LSTM) model – to the problem of predicting student responses to exercises based upon their past activity.

## 3.1 Model

Traditional Recurrent Neural Networks (RNNs) map an input sequence of vectors $\mathbf{x}_1, \ldots, \mathbf{x}_T$, to an output sequence of vectors $\mathbf{y}_1, \ldots, \mathbf{y}_T$. This is achieved by computing a sequence of 'hidden' states $\mathbf{h}_1, \ldots, \mathbf{h}_T$ which can be viewed as successive encodings of relevant information from past observations that will be useful for future predictions. See Figure 2 for a cartoon illustration. The variables are related using a simple network defined by the equations:

$$\mathbf{h}_t = \tanh\left(\mathbf{W}_{hx}\mathbf{x}_t + \mathbf{W}_{hh}\mathbf{h}_{t-1} + \mathbf{b}_h\right), \tag{1}$$

$$\mathbf{y}_t = \sigma\left(\mathbf{W}_{yh}\mathbf{h}_t + \mathbf{b}_y\right), \tag{2}$$

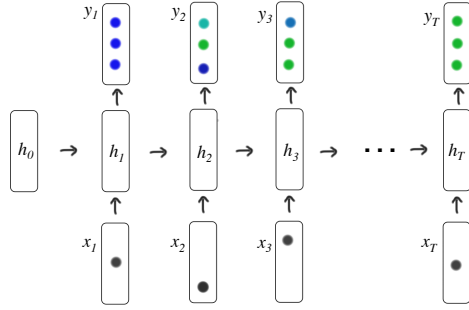

*Figure 2: The connection between variables in a simple recurrent neural network. The inputs ($\mathbf{x}_t$) to the dynamic network are either one-hot encodings or compressed representations of a student action, and the prediction ($\mathbf{y}_t$) is a vector representing the probability of getting each of the dataset exercises correct.*

where both $\tanh$ and the sigmoid function, $\sigma(\cdot)$, are applied elementwise. The model is parameterized by an input weight matrix $\mathbf{W}_{hx}$, recurrent weight matrix $\mathbf{W}_{hh}$, initial state $\mathbf{h}_0$, and readout weight matrix $\mathbf{W}_{yh}$. Biases for latent and readout units are given by $\mathbf{b}_h$ and $\mathbf{b}_y$.

Long Short Term Memory (LSTM) networks [16] are a more complex variant of RNNs that often prove more powerful. In LSTMs latent units retain their values until explicitly cleared by the action of a 'forget gate'. They thus more naturally retain information for many time steps, which is believed to make them easier to train. Additionally, hidden units are updated using multiplicative interactions, and they can thus perform more complicated transformations for the same number of latent units. The update equations for an LSTM are significantly more complicated than for an RNN, and can be found in Appendix A.

## 3.2 Input and Output Time Series

In order to train an RNN or LSTM on student interactions, it is necessary to convert those interactions into a sequence of fixed length input vectors $\mathbf{x}_t$. We do this using two methods depending on the nature of those interactions:

For datasets with a small number $M$ of unique exercises, we set $x_t$ to be a one-hot encoding of the student interaction tuple $h_t = \{q_t, a_t\}$ that represents the combination of which exercise was answered and if the exercise was answered correctly, so $x_t \in \{0, 1\}^{2M}$. We found that having separate representations for $q_t$ and $a_t$ degraded performance.

For large feature spaces, a one-hot encoding can quickly become impractically large. For datasets with a large number of unique exercises, we therefore instead assign a random vector $\mathbf{n}_{q,a} \sim \mathcal{N}(0, \mathbf{I})$ to each input tuple, where $\mathbf{n}_{q,a} \in \mathcal{R}^N$, and $N \ll M$. We then set each input vector $\mathbf{x}_t$ to the corresponding random vector, $\mathbf{x}_t = \mathbf{n}_{q_t, a_t}$. This random low-dimensional representation of a one-hot high-dimensional vector is motivated by compressed sensing. Compressed sensing states that a $k$-sparse signal in $d$ dimensions can be recovered exactly from $k \log d$ random linear projections (up to scaling and additive constants) [2]. Since a one-hot encoding is a 1-sparse signal, the student interaction tuple can be exactly encoded by assigning it to a fixed random Gaussian input vector of length $\sim \log 2M$. Although the current paper deals only with 1-hot vectors, this technique can be extended easily to capture aspects of more complex student interactions in a fixed length vector.

The output $\mathbf{y}_t$ is a vector of length equal to the number of problems, where each entry represents the predicted probability that the student would answer that particular problem correctly. Thus the prediction of $a_{t+1}$ can then be read from the entry in $y_t$ corresponding to $q_{t+1}$.

## 3.3 Optimization

The training objective is the negative log likelihood of the observed sequence of student responses under the model. Let $\delta(q_{t+1})$ be the one-hot encoding of which exercise is answered at time $t + 1$, and let $\ell$ be binary cross entropy. The loss for a given prediction is $\ell(\mathbf{y}^T \delta(q_{t+1}), a_{t+1})$, and the

loss for a single student is:

$$L = \sum_t \ell(\mathbf{y}^T \delta(q_{t+1}), a_{t+1}) \tag{3}$$

This objective was minimized using stochastic gradient descent on minibatches. To prevent over-fitting during training, dropout was applied to $\mathbf{h}_t$ when computing the readout $\mathbf{y}_t$, but not when computing the next hidden state $\mathbf{h}_{t+1}$. We prevent gradients from 'exploding' as we backpropagate through time by truncating the length of gradients whose norm is above a threshold. For all models in this paper we consistently used hidden dimensionality of 200 and a mini-batch size of 100. To facilitate research in DKTs we have published our code and relevant preprocessed data[1].

# 4 Educational Applications

The training objective for knowledge tracing is to predict a student's future performance based on their past activity. This is directly useful – for instance formal testing is no longer necessary if a student's ability undergoes continuous assessment. As explored experimentally in Section 6, the DKT model can also power a number of other advancements.

## 4.1 Improving Curricula

One of the biggest potential impacts of our model is in choosing the best sequence of learning items to present to a student. Given a student with an estimated hidden knowledge state, we can query our RNN to calculate what their expected knowledge state would be if we were to assign them a particular exercise. For instance, in Figure 1 after the student has answered 50 exercises we can test every possible next exercise we could show her and compute her expected knowledge state given that choice. The predicted optimal next problem for this student is to revisit solving for the y-intercept.

We use a trained DKT to test two classic curricula rules from education literature: *mixing* where exercises from different topics are intermixed, and *blocking* where students answer series of exercises of the same type [30]. Since choosing the entire sequence of next exercises so as to maximize predicted accuracy can be phrased as a Markov decision problem we can also evaluate the benefits of using the $\mathrm{expectimax}$ algorithm (see Appendix) to chose an optimal sequence of problems.

## 4.2 Discovering Exercise Relationships

The DKT model can further be applied to the task of discovering latent structure or concepts in the data, a task that is typically performed by human experts. We approached this problem by assigning an influence $J_{ij}$ to every directed pair of exercises $i$ and $j$,

$$J_{ij} = \frac{y(j|i)}{\sum_k y(j|k)}, \tag{4}$$

where $y(j|i)$ is the correctness probability assigned by the RNN to exercise $j$ on the second timestep, given that a student answered exercise $i$ correctly on the first. We show that this characterization of the dependencies captured by the RNN recovers the pre-requisites associated with exercises.

# 5 Datasets

We test the ability to predict student performance on three datasets: simulated data, Khan Academy Data, and the Assistments benchmark dataset. On each dataset we measure area under the curve (AUC). For the non-simulated data we evaluate our results using 5-fold cross validation and in all cases hyper-parameters are learned on training data. We compare the results of Deep Knowledge Tracing to standard BKT and, when possible to optimal variations of BKT. Additionally we compare our results to predictions made by simply calculating the marginal probability of a student getting a particular exercise correct.

| | Overview | | | AUC | | | |
|---|---|---|---|---|---|---|---|
| Dataset | Students | Exercise Tags | Answers | Marginal | BKT | BKT* | DKT |
| Simulated-5 | 4,000 | 50 | 200 K | ? | 0.54 | - | 0.75 |
| Khan Math | 47,495 | 69 | 1,435 K | 0.63 | 0.68 | - | 0.85 |
| Assistments | 15,931 | 124 | 526 K | 0.62 | 0.67 | 0.69 | 0.86 |

*Table 1: AUC results for all datasets tested. BKT is the standard BKT. BKT\* is the best reported result from the literature for Assistments. DKT is the result of using LSTM Deep Knowledge Tracing.*

**Simulated Data**: We simulate virtual students learning virtual concepts and test how well we can predict responses in this controlled setting. For each run of this experiment we generate two thousand students who answer 50 exercises drawn from $k \in 1 \ldots 5$ concepts. For this dataset only, all students answer the same sequence of 50 exercises. Each student has a latent knowledge state "skill" for each concept, and each exercise has both a single concept and a difficulty. The probability of a student getting a exercise with difficulty $\beta$ correct if the student had concept skill $\alpha$ is modelled using classic Item Response Theory [9] as: $p(\text{correct}|\alpha, \beta) = c + \frac{1-c}{1+e^{\beta-\alpha}}$ where $c$ is the probability of a random guess (set to be 0.25). Students "learn" over time via an increase to the concept skill which corresponded to the exercise they answered. To understand how the different models can incorporate unlabelled data, we do *not* provide models with the hidden concept labels (instead the input is simply the exercise index and whether or not the exercise was answered correctly). We evaluate prediction performance on an additional two thousand simulated test students. For each number of concepts we repeat the experiment 20 times with different randomly generated data to evaluate accuracy mean and standard error.

**Khan Academy Data**: We used a sample of anonymized student usage interactions from the eighth grade Common Core curriculum on Khan Academy. The dataset included 1.4 million exercises completed by 47,495 students across 69 different exercise types. It did not contain any personal information. Only the researchers working on this paper had access to this anonymized dataset, and its use was governed by an agreement designed to protect student privacy in accordance with Khan Academy's privacy notice [1]. Khan Academy provides a particularly relevant source of learning data, since students often interact with the site for an extended period of time and for a variety of content, and because students are often self-directed in the topics they work on and in the trajectory they take through material.

**Benchmark Dataset**: In order to understand how our model compared to other models we evaluated models on the Assistments 2009-2010 "skill builder" public benchmark dataset[2]. Assistments is an online tutor that simultaneously teaches and assesses students in grade school mathematics. It is, to the best of our knowledge, the largest publicly available knowledge tracing dataset [11].

## 6 Results

On all three datasets Deep Knowledge Tracing substantially outperformed previous methods. On the Khan dataset using an LSTM neural network model led to an AUC of 0.85 which was a notable improvement over the performance of a standard BKT (AUC = 0.68), especially when compared to the small improvement BKT provided over the marginal baseline (AUC = 0.63). See Table 1 and Figure 3(b). On the Assistments dataset DKT produced a 25% gain over the previous best reported result (AUC = 0.86 and 0.69 respectively) [23]. The gain we report in AUC compared to the marginal baseline (0.24) is more than triple the largest gain achieved on the dataset to date (0.07).

The prediction results from the synthetic dataset provide an interesting demonstration of the capacities of deep knowledge tracing. Both the LSTM and RNN models did as well at predicting student responses as an oracle which had perfect knowledge of all model parameters (and only had to fit the latent student knowledge variables). See Figure 3(a). In order to get accuracy on par with an oracle the models would have to mimic a function that incorporates: latent concepts, the difficulty of each exercise, the prior distributions of student knowledge and the increase in concept skill that happened

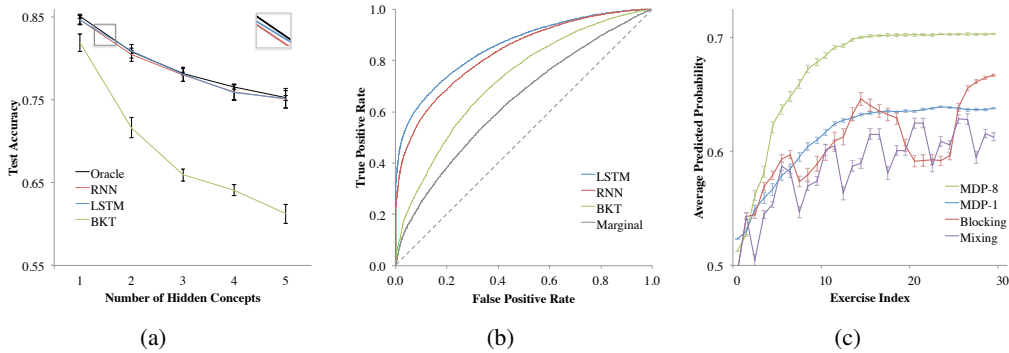

|     |     |     |
| --- | --- | --- |
| (a) | (b) | (c) |

*Figure 3: Left: Prediction results for (a) simulated data and (b) Khan Academy data. Right: (c) Predicted knowledge on Assistments data for different exercise curricula. Error bars are standard error of the mean.*

after each exercise. In contrast, the BKT prediction degraded substantially as the number of hidden concepts increased as it doesn't have a mechanism to learn unlabelled concepts.

We tested our ability to intelligently chose exercises on a subset of five concepts from the Assistment dataset. For each curricula method, we used our DKT model to simulate how a student would answer questions and evaluate how much a student knew after 30 exercises. We repeated student simulations 500 times and measured the average predicted probability of a student getting future questions correct. In the Assistment context the blocking strategy had a notable advantage over mixing. See Figure 3(c). While blocking performs on par with solving expectimax one exercise deep (MDP-1), if we look further into the future when choosing the next problem we come up with curricula where students have higher predicted knowledge after solving fewer problems (MDP-8).

The prediction accuracy on the synthetic dataset suggest that it may be possible to use DKT models to extract the latent structure between the assessments in the dataset. The graph of our model's conditional influences for the synthetic dataset reveals a perfect clustering of the five latent concepts (see Figure 4), with directed edges set using the influence function in Equation 4. An interesting observation is that some of the exercises from the same concept occurred far apart in time. For example, in the synthetic dataset, where node numbers depict sequence, the 5th exercise in the synthetic dataset was from hidden concept 1 and even though it wasn't until the 22nd problem that another problem from the same concept was asked, we were able to learn a strong conditional dependency between the two. We analyzed the Khan dataset using the same technique. The resulting graph is a compelling articulation of how the concepts in the 8th grade Common Core are related to each other (see Figure 4. Node numbers depict exercise tags). We restricted the analysis to ordered pairs of exercises $\{A, B\}$ such that after $A$ appeared, $B$ appeared more than 1% of the time in the remainder of the sequence). To determine if the resulting conditional relationships are a product of obvious underlying trends in the data we compared our results to two baseline measures (1) the transition probabilities of students answering $B$ given they had just answered $A$ and (2) the probability in the dataset (without using a DKT model) of answering $B$ correctly given a student had earlier answered $A$ correctly. Both baseline methods generated discordant graphs, which are shown in the Appendix. While many of the relationships we uncovered may be unsurprising to an education expert their discovery is affirmation that the DKT network learned a coherent model.

# 7 Discussion

In this paper we apply RNNs to the problem of knowledge tracing in education, showing improvement over prior state-of-the-art performance on the Assistments benchmark and Khan dataset. Two particularly interesting novel properties of our new model are that (1) it does not need expert annotations (it can learn concept patterns on its own) and (2) it can operate on any student input that can be vectorized. One disadvantage of RNNs over simple hidden Markov methods is that they require large amounts of training data, and so are well suited to an online education environment, but not a small classroom environment.

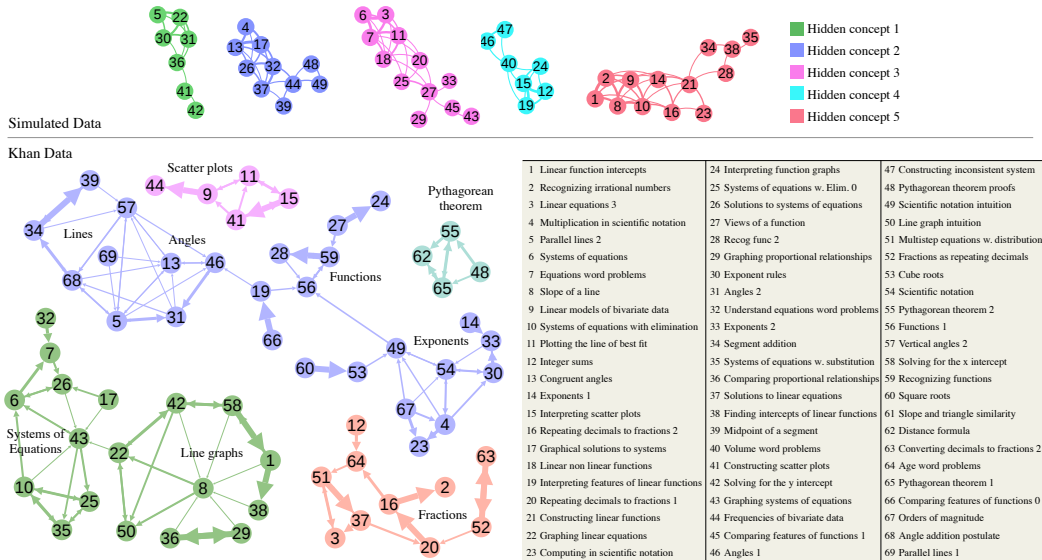

Simulated Data

Khan Data

| | | |
|---|---|---|
| 1 Linear function intercepts | 24 Interpreting function graphs | 47 Constructing inconsistent system |
| 2 Recognizing irrational numbers | 25 Systems of equations w. Elim. 0 | 48 Pythagorean theorem proofs |
| 3 Linear equations 3 | 26 Solutions to systems of equations | 49 Scientific notation intuition |
| 4 Multiplication in scientific notation | 27 Views of a function | 50 Line graph intuition |
| 5 Parallel lines 2 | 28 Recog func 2 | 51 Multistep equations w. distribution |
| 6 Systems of equations | 29 Graphing proportional relationships | 52 Fractions as repeating decimals |
| 7 Equations word problems | 30 Exponent rules | 53 Cube roots |
| 8 Slope of a line | 31 Angles 2 | 54 Scientific notation |
| 9 Linear models of bivariate data | 32 Understand equations word problems | 55 Pythagorean theorem 2 |
| 10 Systems of equations with elimination | 33 Exponents 2 | 56 Functions 1 |
| 11 Plotting the line of best fit | 34 Segment addition | 57 Vertical angles 2 |
| 12 Integer sums | 35 Systems of equations w. substitution | 58 Solving for the x intercept |
| 13 Congruent angles | 36 Comparing proportional relationships | 59 Recognizing functions |
| 14 Exponents 1 | 37 Solutions to linear equations | 60 Square roots |
| 15 Interpreting scatter plots | 38 Finding intercepts of linear functions | 61 Slope and triangle similarity |
| 16 Repeating decimals to fractions 2 | 39 Midpoint of a segment | 62 Distance formula |
| 17 Graphical solutions to systems | 40 Volume word problems | 63 Converting decimals to fractions 2 |
| 18 Linear non linear functions | 41 Constructing scatter plots | 64 Age word problems |
| 19 Interpreting features of linear functions | 42 Solving for the y intercept | 65 Pythagorean theorem 1 |
| 20 Repeating decimals to fractions 1 | 43 Graphing systems of equations | 66 Comparing features of functions 0 |
| 21 Constructing linear functions | 44 Frequencies of bivariate data | 67 Orders of magnitude |
| 22 Graphing linear equations | 45 Comparing features of functions 1 | 68 Angle addition postulate |
| 23 Computing in scientific notation | 46 Angles 1 | 69 Parallel lines 1 |

Figure 4: Graphs of conditional influence between exercises in DKT models. Above: We observe a perfect clustering of latent concepts in the synthetic data. Below: A convincing depiction of how 8th grade math Common Core exercises influence one another. Arrow size indicates connection strength. Note that nodes may be connected in both directions. Edges with a magnitude smaller than 0.1 have been thresholded. Cluster labels are added by hand, but are fully consistent with the exercises in each cluster.

The application of RNNs to knowledge tracing provides many directions for future research. Further investigations could incorporate other features as inputs (such as time taken), explore other educational impacts (such as hint generation, dropout prediction), and validate hypotheses posed in education literature (such as spaced repetition, modeling how students forget). Because DKTs take vector input it should be possible to track knowledge over more complex learning activities. An especially interesting extension is to trace student knowledge as they solve open-ended programming tasks [26, 27]. Using a recently developed method for vectorization of programs [25] we hope to be able to intelligently model student knowledge over time as they learn to program.

In an ongoing collaboration with Khan Academy, we plan to test the efficacy of DKT for curriculum planning in a controlled experiment, by using it to propose exercises on the site.

### Acknowledgments

Many thanks to John Mitchell for his guidance and Khan Academy for its support. Chris Piech is supported by NSF-GRFP grant number DGE-114747.

## Footnotes

[1]https://github.com/chrispiech/DeepKnowledgeTracing

[2]https://sites.google.com/site/assistmentsdata/home/assistment-2009-2010-data

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
