[Supplementary Material]

# Appendix

## A   LSTM Equations

$$\mathbf{i}_t = \sigma(\mathbf{W}_{ix}\mathbf{x}_t + \mathbf{W}_{ih}\mathbf{h}_{t-1} + \mathbf{b}_i) \tag{5}$$
$$\mathbf{g}_t = \sigma(\mathbf{W}_{gx}\mathbf{x}_t + \mathbf{W}_{gh}\mathbf{h}_{t-1} + \mathbf{b}_g) \tag{6}$$
$$\mathbf{f}_t = \sigma(\mathbf{W}_{fx}\mathbf{x}_t + \mathbf{W}_{fh}\mathbf{h}_{t-1} + \mathbf{b}_f) \tag{7}$$
$$\mathbf{o}_t = \sigma(\mathbf{W}_{ox}\mathbf{x}_t + \mathbf{W}_{oh}\mathbf{h}_{t-1} + \mathbf{b}_o) \tag{8}$$
$$\mathbf{h}_t = \mathbf{o}_t \odot \mathbf{m}_t \tag{9}$$
$$\mathbf{m}_t = \mathbf{f}_t \odot \mathbf{m}_{t-1} + \mathbf{i}_t \odot \mathbf{g}_t \tag{10}$$
$$\mathbf{z}_t = \mathbf{W}_{zm}\mathbf{m}_t + \mathbf{b}_z \tag{11}$$
$$\mathbf{y}_t = \sigma(\mathbf{z}_t) \tag{12}$$

## B   Expectimax

Expectimax is a brute force, tree based, MDP search algorithm that calculates the expected utility of each action under the assumption that the agent will always make a maximizing decision when given a choice, and that after an action has been taken, the environment will produce a next state using a stochastic process.

## C   Concept Clustering

*Figure A.1: It is difficult to cluster concepts using model weights. Here is tSNE using the readout and reading weights of the best RNN model trained on synthetic data with five hidden concepts (labeled).*

## D   Model Insights

| | | |
|---|---|---|
| 1 Linear function intercepts | 24 Interpreting function graphs | 47 Constructing inconsistent system |
| 2 Recognizing irrational numbers | 25 Systems of equations w. Elim. 0 | 48 Pythagorean theorem proofs |
| 3 Linear equations 3 | 26 Solutions to systems of equations | 49 Scientific notation intuition |
| 4 Multiplication in scientific notation | 27 Views of a function | 50 Line graph intuition |
| 5 Parallel lines 2 | 28 Recog func 2 | 51 Multistep equations w. distribution |
| 6 Systems of equations | 29 Graphing proportional relationships | 52 Fractions as repeating decimals |
| 7 Equations word problems | 30 Exponent rules | 53 Cube roots |
| 8 Slope of a line | 31 Angles 2 | 54 Scientific notation |
| 9 Linear models of bivariate data | 32 Understand equations word problems | 55 Pythagorean theorem 2 |
| 10 Systems of equations with elimination | 33 Exponents 2 | 56 Functions 1 |
| 11 Plotting the line of best fit | 34 Segment addition | 57 Vertical angles 2 |
| 12 Integer sums | 35 Systems of equations w. substitution | 58 Solving for the x intercept |
| 13 Congruent angles | 36 Comparing proportional relationships | 59 Recognizing functions |
| 14 Exponents 1 | 37 Solutions to linear equations | 60 Square roots |
| 15 Interpreting scatter plots | 38 Finding intercepts of linear functions | 61 Slope and triangle similarity |
| 16 Repeating decimals to fractions 2 | 39 Midpoint of a segment | 62 Distance formula |
| 17 Graphical solutions to systems | 40 Volume word problems | 63 Converting decimals to fractions 2 |
| 18 Linear non linear functions | 41 Constructing scatter plots | 64 Age word problems |
| 19 Interpreting features of linear functions | 42 Solving for the y intercept | 65 Pythagorean theorem 1 |
| 20 Repeating decimals to fractions 1 | 43 Graphing systems of equations | 66 Comparing features of functions 0 |
| 21 Constructing linear functions | 44 Frequencies of bivariate data | 67 Orders of magnitude |
| 22 Graphing linear equations | 45 Comparing features of functions 1 | 68 Angle addition postulate |
| 23 Computing in scientific notation | 46 Angles 1 | 69 Parallel lines 1 |

*Figure A.2: The Khan Academy exercise labels.*

*Figure A.3: Exercise influence graph derived from student transitions between problems. Edges $(a, b)$ represent the probability of a student solving $b$ after they solve $a$. Only transitions with probability $> 0.1$ are displayed. These have less structure than the relationships derived in Figure. 4.*

*Figure A.4: Exercise influence graph using Equation 4, but based on the empirical conditional accuracy on exercise $j$ following correct performance on exercise $i$. Only edge weights $> 0.1$ are displayed. These have less structure than the relationships derived in Figure 4.*

*Figure A.5: How do the best students differ from below-average students? There seems to be much less variance in their knowledge increase. The red curve shows the mean predicted accuracy for students closest to the 40th percentile of the class after 50 questions, while the blue curve is for students closest to the 100th percentile of the class after 50 questions.*

*Figure A.6: The parameter $\mathbf{b}_z$ is easy to interpret. In general the $i$th element captures the marginal probability of getting the $i$th exercise correct.*