[Reviews · NeurIPS 2015]

Submitted by Assigned_Reviewer_1

This article proposes to model student learning in the context of MOOC using RNN. The system is evaluated according to its ability to predict the exercices on which a given student will succeed or not. The authors present a LSTM RNN architecture to tackle this problem and demonstrate the performance improvement over BKT standard method.

This article is an well documented application paper on knowledge tracing, where LSTM RNN seems to be a great advance. Dataset & baselines are relevant and the authors give their code in supplementary material: this effort should be remarked. However, it would have been necessary to add a small README to make the code usable easily.

The organization of the paper should be cleaned: the definition of knowledge tracing in subsection 1.1 is alone and it should not be part of the introduction. On top of that, section 4 is strange: it should be split and integrated in the introduction and the related work.

I do not understand the motivation is using AUC as evaluation metrics: why not relying on accuracy? This choice is not motivated and does not seem to be the reference in knowledge tracing (even if I am not an expert of the field and I saw that Fig 3 presents also accuracy).

Sec 6.2 (and supplementary material) is interesting. Could you imagine a way to build a quantitative evaluation of this task?

In conclusion, I found this article interesting and well written. The breakthrough in performance should justify a publication in NIPS.
Summary: I found this article interesting and well written. The breakthrough in performance should justify a publication in NIPS.

Submitted by Assigned_Reviewer_2

Authors apply LSTM recurrent neural networks to the task of modeling student learning in online courses.

Results are better than some previous baselines.

The paper is written quite well, uses a spectacular dataset in evaluation (47,000 students solving problems on Khan Academy), and gets results that are significantly better than the baselines to which the paper was compared.
Summary: Paper is well-written, and with solid performance results.

Definitely an applications paper of the style, "we applied LSTM's to this problem, using a very large dataset, and got great results."

Submitted by Assigned_Reviewer_3

This paper describes the application of RNNs (simple and LSTM) to the task of predicting student performance based on their previous performance. The improvement over the baseline method (BKT) is very large. There is a brief experiment on improving curricula, and an exploration of the relationships between exercises.

The method is easy to follow, both because the paper is well written (but needs proofreading, see below) and the techniques are not very complicated. Personally, I think that "not complicated" is a good thing, but for some, this might be considered a negative. The paper is a novel application of an existing technique to an existing problem. As mentioned already, the improvement is very large and should be significant for the computer-assisted education field.
Summary: A clear paper with good results on an application in computer-assisted education; from a machine-learning point of view, maybe not as impactful.

Author Feedback
Author rebuttal: We thank the reviewers for their thoughtful reviews, helpful suggestions, and the consistent feedback that the ideas are well presented and the results demonstrate a significant advance.

Overall

We believe that our presentation of a novel application is well suited to NIPS. NIPS has a culture and history of pushing forward both theory and application, and each makes the other stronger. Indeed, the NIPS call for papers specifically cites applications as one of the 10 technical areas of interest. As one recent example, Krizhevsky et al 2012 focused largely on one application, but has been transformative to the fields of computer vision and deep learning. We believe human learning is an exciting target domain that will drive future developments in machine learning.

Moreover, in the process of modelling student learning we were innovative in our use and analysis of RNNs. We have modified our paper to emphasize that:
(1) The insight to allow for a multiplicative interaction in input between the exercise a student works on and their response (as opposed to using input that simply appends the two) was crucial to an RNN working in this domain. Similarly the choice to use compressed sensing on the inputs will be interesting to practitioners of RNNs.
(2) To better understand the evolution of student knowledge we performed novel analyses on the RNN. These included discovering relationships between topics (i.e. unsupervised extraction of mathematical concepts) and the development of optimal curricula. In ongoing work we are further analyzing RNN network dynamics by relating student forgetting to decay towards RNN fixed points.

AI-empowered education has the potential to have extraordinary social impact -- providing personalized high quality education to billions of people. We are excited that the reviewers noted our "impressive gains" in this domain and referred to them as a "breakthrough".

Reviewer 1

We have received a large show of excitement regarding our github code and we are committed to supporting the many people who want to build on what we have started. We will release a documented version of the code (with README) alongside the final paper.

We redistributed and polished the content in sections 1.1 and 4, to improve clarity.

There is often less tolerance for false positives than for false negatives when interacting with students. For this reason, AUC provides relevant information that is missing from classification accuracy alone (and is commonly used in knowledge tracing papers). We now include classification accuracy results in the writeup to show the correspondence.

It is difficult to make a quantitative measure of the accuracy of the topic dependency model, since there is no ground truth data. The true dependencies between exercises in student brains are unknown.

Reviewer 4

We believe that theory and application go hand in hand in computer science, and that introducing new applications can have as large an impact as introducing new theory. Online education has the potential to be an extraordinarily impactful application.

While the social benefit is the dominant story of our paper, the way in which we adapted RNNs to apply to this new domain (and explored the resulting model) constitutes a novel contribution and we believe will be interesting to the deep learning community (see the Overview). The pursuit of a model that can capture and explain human learning is also an interesting path that could help the NIPS community push the state of the art in artificial intelligence. By understanding the dynamics of human learning we can perhaps improve machine learning.

Reviewer 6

We use a one-hot encoding of each possible {exercise, correctness} tuple. E.g., if there were two exercises we would have a one-hot vector with entries indicating [ex1-correct, ex1-incorrect, ex2-correct, ex2-incorrect]. This was an important decision because it allows the RNN to much more effectively capture the interaction between what was answered and whether the answer was correct. We have clarified this in the text.

We represent predicted accuracies as a vector over all exercises in order to maintain a standard notation (and a standard recurrent network implementation). We only evaluate a loss for the target exercise, as you suggest. We have made this explicit in the text.

We have made the specific suggested textual changes. We also worked to clearly structure long sentences by using commas, or by splitting them into multiple sentences.

We now provide an extended description of our curriculum improvement experiment, and of Expectimax. Expectimax here corresponds to the maximization of student expected accuracy after some number of time steps, using a brute-force tree search over possible curricula.

We believe the novel techniques for applying and analyzing RNNs will be of academic interest to the general community in addition to having broad impact.